# An Intelligent Diabetic Patient Tracking System Based on Machine Learning for E-Health Applications

**DOI:** 10.3390/s23063004

**Published:** 2023-03-10

**Authors:** Sindhu P. Menon, Prashant Kumar Shukla, Priyanka Sethi, Areej Alasiry, Mehrez Marzougui, M. Turki-Hadj Alouane, Arfat Ahmad Khan

**Affiliations:** 1School of Computing and Information Technology, Reva University, Bangalore 560064, Karnataka, India; 2Department of Computer Science and Engineering, Koneru Lakshmaiah Education Foundation, Vaddeswaram, Guntur 522302, Andhra Pradesh, India; 3Department of Physiotherapy, Faculty of Allied Health Sciences, Manav Rachna International Institute of Research & Studies, Faridabad 121004, Haryana, India; 4College of Computer Science, King Khalid University, Abha 61413, Saudi Arabia; 5Department of Computer Science, College of Computing, Khon Kaen University, Khon Kaen 40002, Thailand

**Keywords:** e-health, machine learning (ML), Internet of Things (IoT), diabetic patient monitoring, advanced-spatial-vector-based Random Forest (ASV-RF), particle swarm optimization (PSO)

## Abstract

Background: Continuous surveillance helps people with diabetes live better lives. A wide range of technologies, including the Internet of Things (IoT), modern communications, and artificial intelligence (AI), can assist in lowering the expense of health services. Due to numerous communication systems, it is now possible to provide customized and distant healthcare. Main problem: Healthcare data grows daily, making storage and processing challenging. We provide intelligent healthcare structures for smart e-health apps to solve the aforesaid problem. The 5G network must offer advanced healthcare services to meet important requirements like large bandwidth and excellent energy efficacy. Methodology: This research suggested an intelligent system for diabetic patient tracking based on machine learning (ML). The architectural components comprised smartphones, sensors, and smart devices, to gather body dimensions. Then, the preprocessed data is normalized using the normalization procedure. To extract features, we use linear discriminant analysis (LDA). To establish a diagnosis, the intelligent system conducted data classification utilizing the suggested advanced-spatial-vector-based Random Forest (ASV-RF) in conjunction with particle swarm optimization (PSO). Results: Compared to other techniques, the simulation’s outcomes demonstrate that the suggested approach offers greater accuracy.

## 1. Introduction

The healthcare industry is constantly growing and thus provides a wide range of research challenges in the field of computer science. Advances in information and communication technology (ICT), sensors, Big Data analysis, machine learning (ML), and artificial intelligence (AI), all can be employed to meet these challenges. For instance, users of IoT-enabled signal surveillance systems can forecast health related conditions including heart attacks, and chronic fevers. Consequently, this facilitates elder care, as well as senior assistance, wellness, and preventive measures [1]. Providing dependable assistance when necessary and decreasing the patient travel issue can improve the quality of care. The primary purpose of new technology is to continuously monitor patients with prolonged diseases, whose prevalence has grown in current centuries [2]. IoT technology, therefore, offers novel options for diabetic patients. The suggested IoT-based healthcare system combines the ML technique with an advanced sensor system to gather a crucial human physiological signal. The centralized cloud processor has received the gathered signal across wireless media for processing and visualization.

Chronic illnesses are recognized by their protracted duration and need for ongoing care. Patients with prolonged diseases frequently stay for prolonged periods in the hospital for regular monitoring. Diabetes, cancer, and heart disease are a few instances of frequent chronic diseases. Since it causes many deaths each year, diabetes is currently a very serious condition [2]. To have a normal life, the diabetes patient must be under constant surveillance. Diabetes is a long-term illness caused by pancreatic dysfunction, which manifests as either inadequate insulin production or improper insulin utilization by the body [3]. Numerous organs, including the neurons, eyes, and blood vessels, can become dysfunctional or deteriorate due to abnormally high or low sugar levels. As a result, regular and ongoing intensive care is essential to prevent the condition of diabetes patients from getting worse [4].

Due to the rise in diabetic patients over the past few years, more technologies are needed to monitor these people. Blood glucose levels are routinely monitored via monitoring equipment for people with diabetes [5]. Consequently, diabetics, family members, and medical professionals can constantly monitor the glucose readings and respond swiftly when there is an alarming reading. The benefit of using portable surveillance systems for diabetes patients is that they spend less time in the hospital, which improves their quality of life [6]. This makes it quite intriguing to employ wireless connectivity with excellent coverage to convey patient information to doctors. In this way, 5G technology—the next era of wireless services for high-speed communication, increased network bandwidth, and network flexibility. Nevertheless, the assessment of this technique is presently concentrated on speeding up data transport [7].

Numerous research papers [8,9,10,11,12,13,14,15,16,17,18,19,20] published recently have suggested various intelligent patient tracking systems, but they have not given enough consideration to accurate, timely assistance or measured parameter delay-associated parameters during transmission. Consequently, an IoT-based health service employing ML algorithm is provided ahead of time to address medical emergencies in real time. In this study, we created a data categorization ML algorithm-based framework for the intelligent regular inspection of people with diabetes. This work’s main goal is to apply ML for forecasting. IoT connects sensors to collect continuous time series information and execute cloud-based data processing.

The paper’s structure is as follows: Part 2—literature survey; Part 3—methodology; Part 4—experimental findings; Part 5—conclusion. 

The contribution of this work lies in the following:An intelligent method for monitoring diabetes patients that uses ML.The architectural components include smart gadgets, sensors, and cellphones, all used to obtain body measurements.The normalization approach is then used to normalize the pre-processed data. Linear discriminant analysis (LDA) is used to extract features.The intelligent system performed data categorization using the proposed advanced-spatial-vector based Random Forest (ASV-RF) together with particle swarm optimization (PSO) to generate a diagnosis.

## 2. Literature Survey

In this section, a wide range of studies on the health monitoring system for diabetes patients are presented.

“An adaptive and predictive context-aware monitoring system” was presented by the authors of [8] as a solution to the problems of continuous monitoring, a shortage of abnormality diagnosis methods, and prediction approaches needing lengthy training periods. An outline of current technology trends for developing a system for HIoT data protection is given in [9], and the accompanying security issues are then examined. Additionally, they offer a structural platform for tracking the health indicators of patients with disabilities or chronic degenerative diseases. Various use-case scenarios illustrate how application components interact with one another. One of the most common medical problems in daily life is the diagnosis and prognosis of diabetes. Consistency and other factors contribute to the body’s long-term micro-vessel problems of diabetes. The experimental test was conducted systematically in [10] utilizing a variety of machines. Understand the classifiers to estimate the prevalence of type-A diabetes in people. The authors of [11] presented a “smart healthcare recommendation system for multidisciplinary diabetes disease patients (SHRS-M3DP)” model to predict the disease quickly and accurately in the patients. However, a better generalized, efficient diagnosis and suggestion approach for various human illnesses still must be developed.

A comprehensive overview of pervasive, intelligent, and networked health services for tracking individuals having chronic and lifestyle illnesses was presented in [12]. The intelligent patient tracking and management architecture employs deep learning (DL) and cloud-based analyses. Another approach was presented in [13], where a support vector machine (SVM) was used to forecast diabetes probability. The samples of females of Pima Indian heritage were only used in the database, which introduced bias based on ethnicity and gender. The fundamental relationship may reduce generalizability even though these variables were not chosen as characteristics in the feature selection process when training the system. Consequently, this approach was only evaluated with two-woman helpers and one clinical examiner, and the outcomes were as anticipated.

The study of [14] focuses on real-time data for improved prediction and accuracy utilizing ML and IoT. The suggested hardware and software system aids patients in early cardiac disease prediction. According to the actual demands and difficulties that the elders and their caretakers face, novel medical treatments are contrasted in [15]. A systematic review of methods for diabetes mellitus identification, detection, and self-management is conducted by [16]. Although it included the main research contributions from 2015 to 2020 in the field, it has missed some of the pertinent contributions appearing in the following years until now. 

The authors of [17] presented a brand-new health-monitoring mechanism to record the disease burden by forecasting illnesses based on primary data gathered from individuals who are accessible in remote locations. Additionally, they suggest a safe data storage architecture for protecting patient data in cloud systems. Here, they provide two brand-new cryptographic techniques for encrypting and decrypting data. To illustrate and assess the software system’s efficacy in managing diabetes, the paper [18] outlines the design and development of a programming system to enhance treatment adherence using an ML technique. The numerous aspects that have an impact on diabetes patients’ health are addressed by the suggested method for this control system [21]. The current system records the user’s walking activity and stores the route, but it does not link the route recorded to the number of calories burned. The author of [22] mentions that a literature review of the work has been done, focusing on the benefits of merging telemedicine with AI. These advantages present endless growth opportunities. The article also examines how AI and telemedicine have been utilized to enhance continuous monitoring and the challenges these methods are intended to solve. The author of [23] uses the patient’s glucose and blood pressure values, the aim is to forecast their hypertension and diabetes state. Classification methods for supervised machine learning are used. In this case, a system is taught to forecast the patient’s blood pressure and diabetes state. The support vector machine classification method was deemed the most accurate after evaluating all the classification algorithms and was therefore selected to train the model. The study [24] provides a model that uses light transmission to calculate the amount of glucose in the body. As the Li-Fi technology is both quicker and more efficient than the conventional Wi-Fi networks, it is the one that is employed. According to the author of [25], IoT and AI are employed to investigate the healthcare industries in order to enhance patient support and patient care going forward. The precision of the patient support process is decreased by the inability of traditional healthcare aid systems to anticipate the precise patient health information and demands. A patient’s personal specifics, such as their medical reports, temperature, fitness tracker, body mass, health activities, and other health care information, are properly predicted using an IoTs sensor with AI, which aids in choosing the best help procedure. The study [26] introduced grey filter bayesian convolution neural network (GFB-CNN), a real-time data- and deep-neural-network-driven IoT smart healthcare strategy. They proposed a GFB-CNN-based, AI-driven Internet of Things (IoT) eHealth architecture to enhance precision and efficiency across the essential quality of service parameters. The article [27] begins with a discussion of the technologies involved in the design of 5G e-health systems from the physical layer, the application layer, and the cross-layer viewpoint. Table 1 depicts the literature survey.

## 3. Methodology

The preprocessing, extraction of features, and classification processes listed in Figure 1 are the basic processes in this section’s construction of a classification utilizing ML.

### 3.1. Dataset

The records of 62 people with diabetes (44 males and 18 females), who endured 67 days of examinations on average, were incorporated into the dataset for this investigation [19]. The glucose concentration set consists of 12,612 glucose concentration datapoints and 5 characteristics.

### 3.2. Preprocessing

The key aspect before using ML algorithms is data preprocessing. Because actual data are frequently noisy, insufficient, and unreliable, they cannot be used immediately in the prediction step. To adequately describe the evidence for the diagnosis of diabetes disease, a preprocessing stage is used.

The diabetes disease datapoint $D^{d}$ has a variety of characteristics; each characteristic has a unique set of numeric numbers, which makes processing more challenging. As a result, a normalization system is employed to normalize the datapoint $D^{d}$ in the range from 0 to 1 and reduce the numerical burden of the diabetes disease prognosis computation. Data normalization can be achieved using a variety of techniques. A min–max normalizing method is applied in the suggested system. Using Equation (1), this approach displays a quantitative score from the given dataset into$\;D_{nor}$ with range [0, 1]:(1)$$D_{nor}=\frac{D^{d}-D_{min}}{D_{max}-D_{min}}\times \left[new\_max-new\_min\right]+new\_min$$

Here, ne=1 and ne=0 is used. Using this approach, the values of all the characteristics fall inside the range from 0 to 1.

### 3.3. Feature Extraction Using LDA

One of the main challenges in ML is the extraction of features, which is a crucial step. By combining the previous dimensions, feature extraction develops new dimensions.

Linear discriminant analysis (LDA) is a form of class-based discrimination. This method helps supervised learning discover a collection of basis vectors. These fundamental vectors are shown as $w_{k}$. The proportion of the between and within class disperses from the training instance set—which is maximized—makes up the $w_{k}$ vectors. The following generalized eigenvalue issue is resolved for discovering $w_{k}$ basis vectors.
(2)$$W_{opt}=\begin{array}{c} argmax\\ w \end{array}\frac{\left|W^{T}S_{C}W\right|}{\left|W^{T}S_{v}W\right|}=\left[w_{1},w_{2},\ldots ,w_{L}\right]$$

Here, $L=dubspace^{\prime }sdimension$, $S_{C}=between\;and\;S_{v}=withinclasses$
(3)$$S_{C}=\overset{a}{\underset{k=1}{\sum }}M_{k}\left(\mu_{k}-\mu \right){\left(\mu_{k}-\mu \right)}^{T}$$
(4)$$S_{V}=\overset{a}{\underset{k=1}{\sum }}\underset{x_{u}\in X_{k}}{\sum }\left(x_{u}-\mu_{k}\right){\left(x_{u}-\mu_{k}\right)}^{T}$$

Here, a=no. ofclass, $X\in R^{N}=sample$, $X_{k}=sampleset$, $M_{k}=no.\;ofclassink$, and μ=mean.

First L greatest eigenvalues $\left\{\psi_{k}|1\leq k\leq L\right\}$ are the base vectors $w_{k}$ desired in Equation (1) if $S_{V}$ is not singular. Attributed to the reason that the LDA base vectors were orthogonal to one another, it may be projected using a basic linear method, $W^{T}x$, into the LDA subspace to derive its representations.

### 3.4. Classification Using ASV-RF Algorithm

Generally speaking, Random Forest (RF) significantly outperforms the single tree classifier. It offers an effective method for categorizing sets of sparse data. However, because basic RF chooses features at random, it is simple to choose irrelevant or distracting characteristics, mainly when the training data is noisy. This could produce subpar categorization outcomes. As explained in earlier sections, the data matrix for type categorization has numerous missing values, which adds noise. It must be improved to use the basic RF in search form categorization.

Because to the shallow feature space, there are a lot of missing values in the training data set. Therefore, many more missing data points cause characteristics to lose importance or possibly become noisy. An unreliable classification tree will emerge from the randomized feature selection for bootstrap samples, which may yield many irrelevant or noisy features. Creating a feature weighting method for building a high-quality classifier seems attractive. By using a weighting system during feature selection as opposed to random selection, we expanded the basic RF. The weighting metrics are set to be chi-squared and are represented as Equation (5).
(5)$$\;\;\;\chi^{2}=\overset{m}{\underset{i=1}{\sum }}\overset{2}{\underset{j-1}{\sum }}\frac{{\left(O_{ij}-e_{ij}\right)}^{2}}{e_{ij}}$$

Here, the definition of $O_{ij}$ as a measured value, which denotes the number of a joint incident, is Equation (5).
(6)$$O_{ij}=count\left(A=a_{i}\cap C=c_{j}\right)$$

Similarly,
(7)$$expected\;value=e_{ij}=\frac{count\left(A=a_{i}\right)\times count\left(C=c_{i}\right)}{N}$$

Equation (4) calculates a weight for every characteristic in the feature space, while only the characteristics with high weights are considered to construct the decision tree (DT).

We constructed a collection of decision trees and then combined the output of each classifier using a likelihood estimation method. Suppose that the input case x is the testing case and that every classification model (DT) $h_{j}\left[j=1\ldots k\right]$ chooses for the potential target class $c_{i}$. Every classifier’s output can be estimated as $P(I=|h_{j})$. The final categorization results are then calculated by adding the probability values as Equation (7):(8)$$P\left(I\left(x\right)=c_{i}\right)=\frac{1}{k}\overset{k}{\underset{j=1}{\sum }}P(I\left(x\right)=c_{i}|h_{j})$$

X input vector belongs to $c_{i}$ if and only if $c_{i}$ has the highest likelihood. Algorithm 1 is a representation of the ASV-RF.

Assuming that there are n features, $\left[\beta .n\right]$ features would be chosen as the training set in stage 3, wherein β represents the feature selection frequency. Learning separate classifiers from the training data supplied is stage 4. The bootstrapping approach is used to choose training data. Sampling with the replacement is employed to choose *t* features from *n*’ features (Here, $t=\left[\log_{2}n+1\right]$). Equation (7) is employed to categorize the unlabeled cases, and after every round, the trained DT is inserted to the forest M*.
**Algorithm 1:** ASV-RF’s procedureInput: *d* = data; *n* = feature; *c* = target class; *k* = no. of DTs; *β* = frequency of selecting features.Result: M* = decision forest.Stage 1: Estimate weights using Equation (4)Stage 2: Sorting features as per weight (W) in decreasing order;$\mathrm{Stage}\;3:\;\mathrm{Allowing}\;n’\;\mathrm{to}\;\mathrm{equal}\;\left[\beta .n\right]$$,\;\mathrm{choose}\;\left[\beta .n\right]$ features with higher Ws as training sets.Stage 4: for i=1 to k doThe bootstrapping approach is used to choose training data (*d*’).When t features are chosen at random, the choice is biased in favor of features with high Ws.Create a C4.5 DT using the *d*’ data and chosen features.Trained DT is added to M*.End forStage 5: Perform classification with M* relying on Equation (7).

### 3.5. PSO

Fish schooling and bird flocking are the sources of inspiration for particle swarm optimization (PSO). A community of particles is created whose present location provides the cost function that needs to be decreased to get the optimal result in a multidimensional space. After every iteration, the advanced velocity and position of the individual particles are revised based on an averaged impact of the current velocity, distance from its best showing so far during the search phase, as well as the distance from the foremost particle, i.e., the particle generating the better outcome so far.

In a multidimensional solution space, a particle’s location and velocity are often represented by the variables *x* and *v*. The d×1 vectors $=(x_{i1},x_{i2,\ldots }x_{id}$) and $=(v_{i1},v_{i2,\ldots }v_{id}$) represent the position and velocity of a particle in d-dimensional space, correspondingly. For each particle *i*, the better location exposed so far is noted as another d×1 vector$\;pbest_{i1,}pbest_{i2,}\ldots \ldots pbest_{id}$. The best global particle among all particles *i* is represented as gbest, and its position in the *d*th dimension is $gbest_{d,}$. Based on the performance of the *k*th iteration, the velocity, and position updating formulas for the particle in the *d*th dimensions in the (*k* + 1)th iteration are as follows:(9)$$\begin{array}{cc} v_{id}^{\left(k+1\right)}=w\times v_{id}^{k} & \;+c_{1}\times rand\times \left(pbest_{id,}-x_{id}^{k}\right)\\ & \;+c_{2}\times rand()\times \left(gbest_{d,}-x_{id}^{k}\right) \end{array}$$
(10)$$x_{id}^{\left(k+1\right)}=x_{id}^{k}+v_{id}^{\left(k+1\right)},\;i\in Np,\;d\in D$$

Here, *D* represents a dimension in the multi-dimensional search issue, and *N_P_* represents the size of the population. *c*_1_ and *c*_2_ are acceleration constants that provide proportional random weight of the deviation from the best individual performance of the particles and the best collective efficiency in the *d*th dimension.

To balance global and local investigations well, the proposed system applies the PSO with an adjustable inertia weight, *w*, during the whole search procedure. According to the following equation, the moment of inertia w is determined in this study.
(11)$$w=w_{max}-\frac{w_{max}-w_{min}}{iter_{max}}\times ite$$

Here, $iter_{max}$ is the maximum number of iterations, and iter is the present number of iterations. We begin with a large value of $w_{max}$ so that we may run an aggressive global search in search of a possible good solution and progressively lower *w* so that we can fine-tune our search locally as we grow closer and closer to the minimal point. 

## 4. Experimental Findings

Here, we test our proposed ASV-RF method regarding the diabetic patient monitoring system. This experiment is carried out using Python, and the collected data samples are used to perform the tests. Our proposed algorithm is also compared with existing algorithms (Sequential minimal optimization (SMO) [20], SVM [21], and DT [21]), to gain our method with the maximum performance in expressions of Accuracy, Sensitivity, Precision, Recall, Specificity, F1-score, TP, FP, Kappa, MAE, and RMSE metrics. This research aims to assess the employed techniques’ efficacy and suggest the most effective algorithm for prediction. We assess the prediction outcomes using a variety of evaluation metrics via the confusion matrix. Figure 2 depicts the representation of the confusion matrix. 

Table 2 and Figure 3 represent the results of proper and improper classified data and the training time for both existing and proposed algorithms. Training time is the time taken by an algorithm to train on a dataset. SMO has 0.032 s, SVM has 0.027 s, DT has 0.051 s, and ASV-RF has 0.019 s in terms of training time to train the dataset. Therefore, it is evident that the suggested approach trains the dataset faster than existing methods. Additionally, our proposed method correctly classifies more data than existing methods. That means the proposed method’s improper classification rate is much less when compared with rates from the existing methods.

Similarly, Table 3 depicts the various metrics’ comparative results of proposed and existing methods. The percentage of samples for which the suggested method correctly predicted outcomes is presented as the system’s effectiveness. The accuracy is calculated using Equation (11). It’s a measure of how many samples are correctly categorized. It determines the degree of similarity between the final results and the input data. The graph demonstrates how the new technique is more accurate than the old one.
(12)$$\mathrm{Accuracy}=\frac{\mathrm{TP}+\mathrm{TN}}{\mathrm{TP}+\mathrm{TN}+\mathrm{FP}+\mathrm{FN}}$$

One of the most crucial metrics for accuracy is precision, calculated as the proportion of properly classified cases to all instances of predictively positive data, as shown in Equation (12). It measures the precision of the recommended procedure by comparing the number of actual successes with the number of expected successes. The performance of the suggested technique is evaluated by distinguishing between true and false positives.
(13)$$\mathrm{precision}=\frac{\mathrm{TP}}{\mathrm{TP}+\mathrm{FP}}$$

The ability of the suggested model to recognize each significant sample in a data collection is known as sensitivity. It is determined statistically by dividing the TPs percentage by the total TPs and FNs (Equation (13)).
(14)$$\mathrm{sensitivity}=\frac{\mathrm{TP}}{\mathrm{TP}+\mathrm{FN}}$$

**Table 3 sensors-23-03004-t003:** Comparative assessment of proposed and existing methods.

Methods	SMO [24]	SVM [25]	DT [25]	ASV-RF [Proposed]
TP	97.99%	98.18%	96.13%	99.8%
FP	1.02%	0.71%	0.99%	0.50%
Accuracy	98.22%	98.44%	96.33%	99.86%
Precision	94.63%	97.93%	98.1%	99.61%
Sensitivity	96.81%	98.22%	97.03%	99.13%
Specificity	97.62%	97.43%	96.44%	98.97%
recall	97.94%	97.65%	97.06%	99.97%
F1-score	95.85%	99.05%	95.34%	98.89%

Figure 4 and Figure 5 depict the comparative assessments of various metrics for proposed and existing methods. The proposed model’s recall is the capacity to identify every important sample in a data collection. It is intended statistically as the proportion of the TPs divided by the summation of the TPs and FNs (Equation (14)).
(15)$$\mathrm{Recall}=\frac{\mathrm{FN}}{\mathrm{FN}+\mathrm{TP}}$$

The f1-score incorporates both into a single factor (Equation (15)) by calculating the harmonic mean of the proposed model’s recall and precision. Specificity is the likelihood of a negative outcome under the premise that the result is, in fact, negative. This probability is often referred to as the real negative rate.
(16)$$F1-\mathrm{score}=\frac{\left(\mathrm{precision}\right)\times \left(\mathrm{recall}\right)\times 2}{\mathrm{precision}+\mathrm{recall}}$$

The proportion between the value of TNs and the total amount of TNs and FPs is referred to as specificity (Equation (16)).
(17)$$\mathrm{specificity}=\frac{\mathrm{TN}}{\mathrm{TN}+\mathrm{FP}}$$

From Table 3, SMO has 98.22%, SVM has 98.44%, DT has 96.33%, and ASV-RF has 99.86% in terms of accuracy. SMO has 94.63%, SVM has 97.93%, DT has 98.1%, and ASV-RF has 99.61% in terms of precision. SMO has 96.81%, SVM has 98.22%, DT has 97.03%, and ASV-RF has 99.13% in terms of sensitivity. SMO has 97.94%, SVM has 97.65%, DT has 97.06%, and ASV-RF has 99.97% in terms of recall. SMO has 95.85%, SVM has 99.05%, DT has 95.34%, and ASV-RF has 98.89% in terms of f1-score. SMO has 97.62%, SVM has 97.43%, DT has 96.44%, and ASV-RF has 98.97% in terms of specificity.

Table 4 and Figure 6 depict the outcomes of kappa, MAE, and RMSE metrics for both proposed and existing methods. A measure that contrasts actual accuracy versus predicted accuracy is called Kappa. SMO has 91.92%, SVM has 95.06%, DT has 94.03%, and ASV-RF has 98.52% in terms of Kappa value. MAE estimates the average degree of mistakes in a set of forecasts. All individual differences are equally weighted in the testing sample’s mean of the absolute disparities between predicted and observed. In Table 4 SMO has 3.48%, SVM has 1.19%, DT has 2.08%, and ASV-RF has 1.01% in terms of MAE. RMSE is a metric used to evaluate the reliability of forecasts. SMO has 13.54%, SVM has 8.16%, DT has 9.08%, and ASV-RF has 7.25% in terms of RMSE. As shown, the suggested ASV-RF technique performs better than other methods using these measures.

Providing accurate patient information to the hospital to protect the patient’s life is referred to as “security of life.” A threat to the patient’s health might result from failure to comply. By misusing the devices, people with evil intentions might transmit inaccurate data to the hospital Figure 7 depict the outcomes of security of life. It is observed that SMO has 85%, SVM has 91%, DT has 73%, and ASV-RF has 91% in terms of security of life. This chart demonstrates that the suggested approach of ASV-RF has a high value.

## 5. Conclusions

E-health trackers keep track of a person’s actions and offer useful feedback, especially when dangerous circumstances arise. This paper presented the ASV-RF method for smart patient monitoring. With the help of this technique, it was possible to assess the person’s dependencies, forecast his future health status, and foresee its decline before potential consequences. The normalization approach was used to normalize the raw dataset for further processes regarding patient monitoring. For feature extraction, the LDA method was employed. The study’s findings revealed that the suggested strategy worked superior to other current approaches in terms of accuracy (99.86%), sensitivity (99.13%), precision (99.61%), recall (99.97%), specificity (98.97%), f1-score (98.89%), TP (99.8%), FP (0.5%), Kappa (98.52%), MAE (1.01%), and RMSE (7.25%) metrics. The proposed framework can be extended with various large datasets in the future. Researchers in this healthcare field will benefit from academic study and methods, particularly from computerized forecasting and virtual assistants for human disorders. They want to regularly gather user input in our future development and feature enhancements. It will keep our application focused on the patient, allowing us to consider users’ demands while refining current features and building new ones. Last but not least, we must always protect the privacy of our consumers as our first concern. The application will have openings for a data breach or leak.

## Figures and Tables

**Figure 1 sensors-23-03004-f001:**
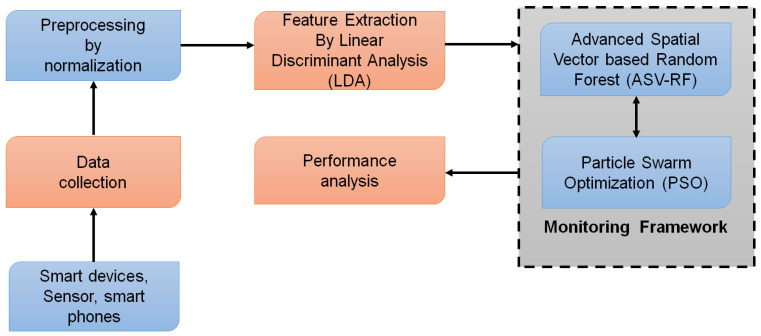
Proposed framework.

**Figure 2 sensors-23-03004-f002:**
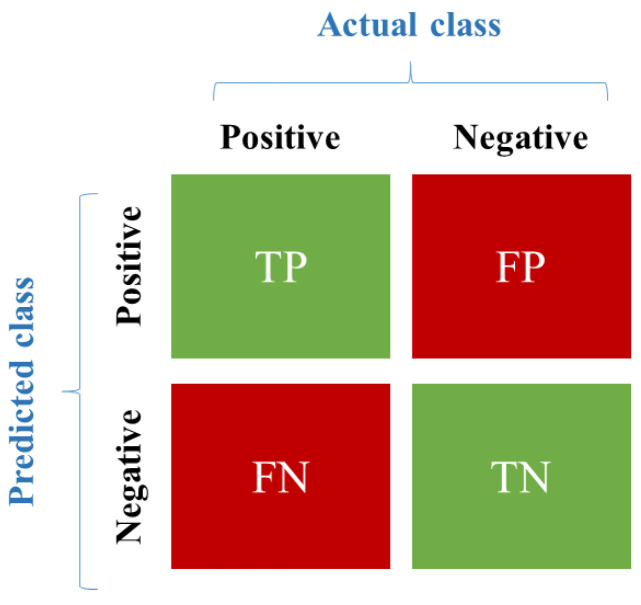
Confusion matrix.

**Figure 3 sensors-23-03004-f003:**
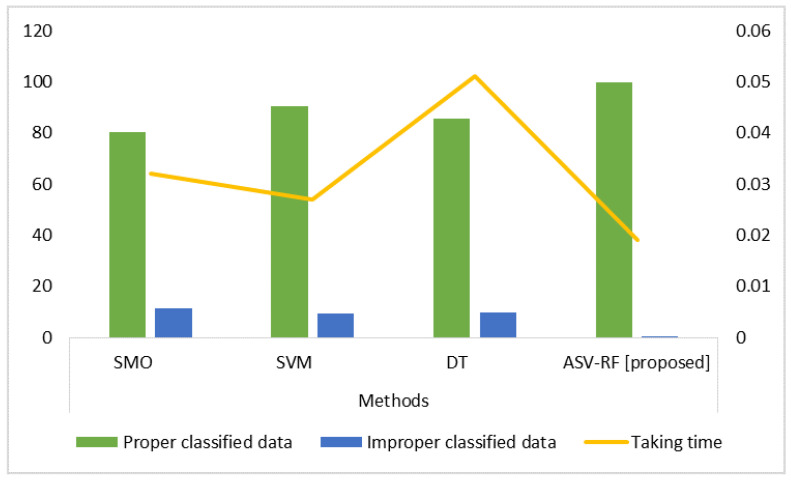
Proper and improper classifications’ results.

**Figure 4 sensors-23-03004-f004:**
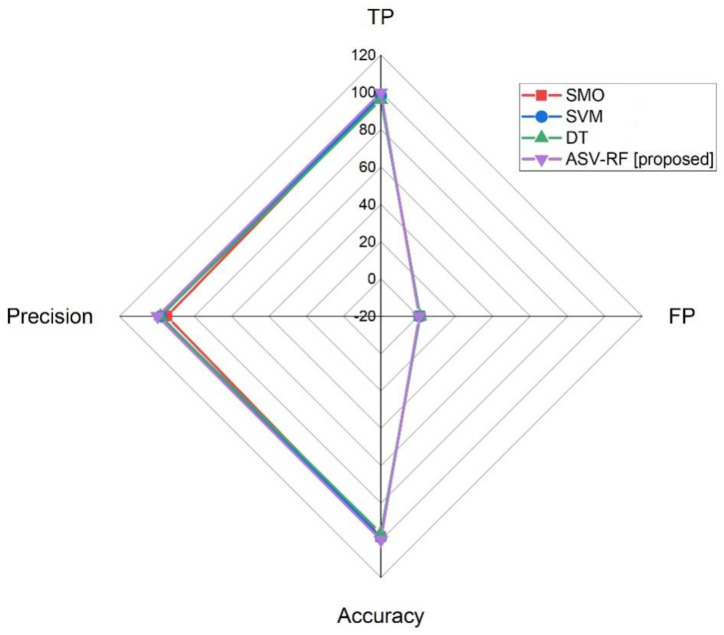
Results of TP, FP, accuracy, and precision metrics.

**Figure 5 sensors-23-03004-f005:**
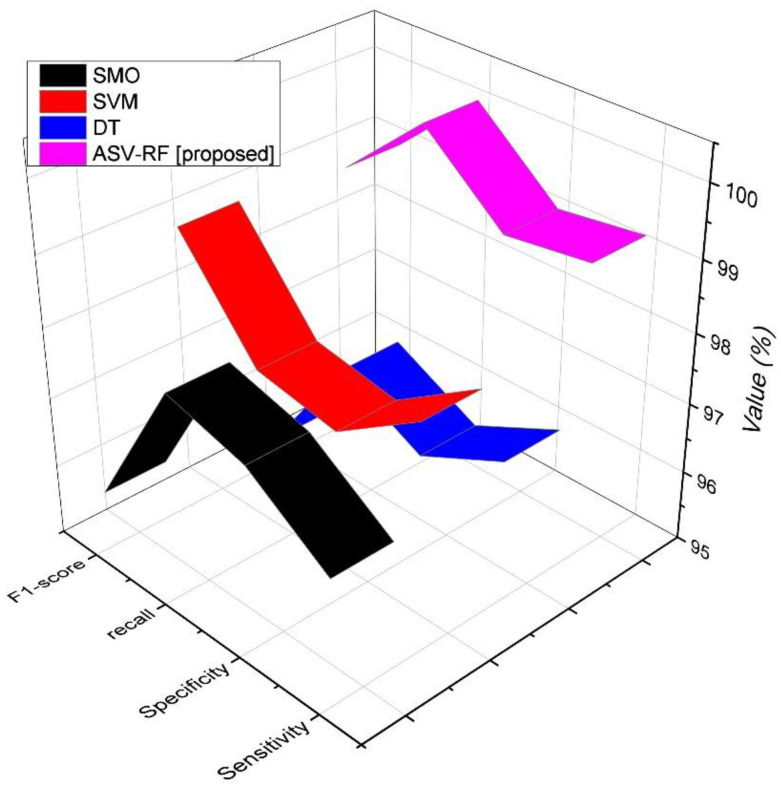
Results of f1-score, recall, sensitivity, and specificity metrics.

**Figure 6 sensors-23-03004-f006:**
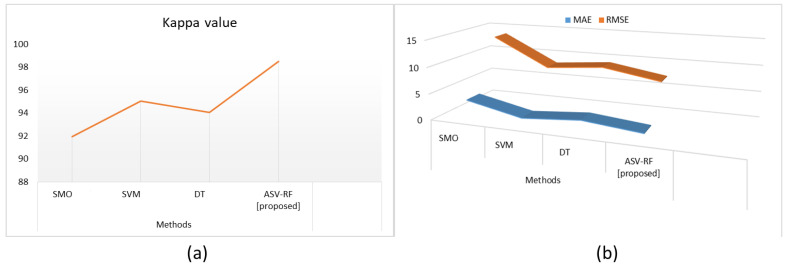
Results of Kappa (**a**), MAE, and RMSE metrics (**b**).

**Figure 7 sensors-23-03004-f007:**
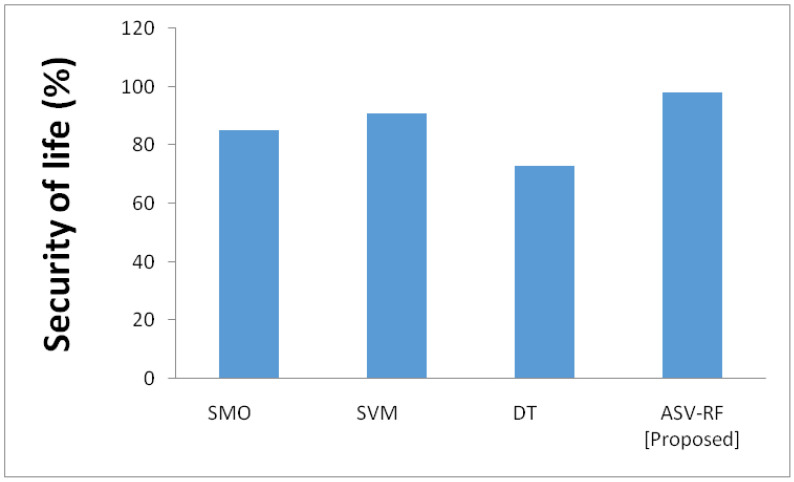
Results of security of life.

**Table 1 sensors-23-03004-t001:** Literature survey.

Reference No.	Title	Author	Algorithm Used	Advantages	Disadvantages
[8]	Adaptive Monitoring System for e-Health Smart Homes	Mshali et al., 2018	An Adaptive Predictive Context-Aware Monitoring System	Provide better results	Time complexity
[11]	A Smart Healthcare Recommendation System for Multidisciplinary Diabetes Patients with Data Fusion Based on Deep Ensemble Learning	Ihnaini et al., 2021	Smart Healthcare Recommendation System for Multidisciplinary Diabetes Patients.	Data fusion allows low-power sensors.	Data conflicts yield unexpected consequences.
[12]	Novel framework based on deep learning and cloud analytics for smart patient monitoring and recommendation	Motwani et al., 2021	Categorical Cross Entropy	Helps assess model correctness	Requires more time in monitoring
[13]	A remote healthcare monitoring framework for diabetes prediction using machine learning	Ramesh et al., 2021	Support Vector Machine Radial Basis Function	SVMs are good at handling high-dimensional data and small datasets.	Unsuitable to Large Datasets. Large training time.
[14]	Smart Health Monitoring System using IOT and Machine Learning Techniques	Pandey et al., 2020	Support Vector Machine	It is robust to outliers	Unsuitable to Large Datasets
[15]	An IoMT-Enabled Smart Healthcare Model to Monitor Elderly People Using Machine Learning Technique	Khan et al., 2021	Smart Healthcare Model	Fastest and most accurate medical treatment	Physical demands
[17]	Cloud- and IoT-based deep learning technique-incorporated secured health monitoring system for dead diseases	Malarvizhi Kumar et al., 2021	Multi-Channel Spatio-Temporal Convolutional Neural Network	Provide a reliable stock price	A lot of training data is needed
[18]	Design and Development of Diabetes Management System Using Machine Learning	Sowah et al., 2020	K-Nearest Neighbour	It can naturally handle multi-class cases	It’s difficult to pick the “correct” value of K

**Table 2 sensors-23-03004-t002:** Proper and improper classified data.

Methods	SMO [20]	SVM [21]	DT [21]	ASV-RF [Proposed]
Proper classified data	80.366%	90.225%	85.652%	99.721%
Improper classified data	11.614%	9.371%	10.025%	0.214%
Taking time	0.032 s	0.027 s	0.051 s	0.019 s

**Table 4 sensors-23-03004-t004:** Results of Kappa, MAE, and RMSE metrics.

Methods	SMO [24]	SVM [25]	DT [25]	ASV-RF [Proposed]
Kappa value	91.92%	95.06%	94.03%	98.52%
MAE	3.48%	1.19%	2.08%	1.01%
RMSE	13.54%	8.16%	9.08%	7.25%

## Data Availability

Not applicable.

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
