# Peer review of "An Intelligent Diabetic Patient Tracking System Based on Machine Learning for E-Health Applications"

_sensors, 2023, doi:10.3390/s23063004_

Round 1

Reviewer 1 Report

This paper can be accepted with minor revisions. 

1. Abstract is too long try to summarize it

2. Explain more primary contribution points at the end of the introduction.

3. Add one table in related work which should be having limitations of previous studies.

4. Separately write future directions at the end of the conclusion.

5.  The security analysis of the proposed work can be included.

6. Make sure that proof-reading is done by the author.

7. Author can include more references related to the topic.

Author Response

                                     Reviewer#1

  1. Abstract is too long try to summarize it

Answer: Thank you for your comment. Based on the comment, we have updated the manuscript.

Background: Continuous surveillance helps persons with diabetes live better lives. A wide range of technologies, such as the Internet of Things (IoT), modern communications, and artificial intelligence (AI), can assist in lowering the expense of health services. Due to numerous communication systems, it is now possible to provide customized and distant healthcare. Main problem: Healthcare data grows daily, making storage and processing challenging. We provide intelligent healthcare structures for smart e-health apps to solve the aforesaid problem. The 5G network must offer advanced healthcare services to meet important requirements like large bandwidth and excellent energy efficiency. Methodology: This research presents an intelligent system for diabetic patient tracking using machine learning (ML). The architectural components comprised smart devices, sensors, and smartphones to gather body measurements. Then, the preprocessed data is normalized using the normalization procedure. To extract features, we use Linear Discriminant Analysis (LDA). To establish a diagnosis, the intelligent system conducted data classification utilizing the suggested Advanced Spatial Vector based Random Forest (ASV-RF) in conjunction with Particle Swarm Optimization (PSO). Results: Compared to other techniques, the results of the simulation show that the suggested approach offers greater accuracy. 

Keywords: e-health, Internet of Things (IoT), machine learning (ML), diabetic patient monitoring, Advanced Spatial Vector based Random Forest (ASV-RF), Particle Swarm Optimization (PSO)

  1. Explain more primary contribution points at the end of the introduction.

Answer: Thank you for your comment. Based on the comment, we have updated the manuscript.

  • The results of this study show an intelligent method for monitoring diabetes patients that uses machine learning (ML).
  • The architectural components include smart gadgets, sensors, and cell phones, all used to obtain body measurements.
  • The normalisation approach is then used to normalise the pre-processed data. Linear Discriminant Analysis is used to extract features (LDA).
  • The intelligent system performed data categorization using the proposed Advanced Spatial Vector based Random Forest (ASV-RF) together with Particle Swarm Optimization (PSO) to generate a diagnosis.
  1. Add one table in related work which should be having limitations of previous studies.

Answer: Thank you for your comment. Based on the comment, we have updated the manuscript.

Table 1: Literature survey

Reference No

Title

Author

Algorithm used

Advantages

Disadvantages

8.

Adaptive Monitoring System for e-Health Smart Homes

Mshali et al. 2018

An Adaptive Predictive Context-Aware Monitoring System

Provide  better results

Time complexity

11.

A Smart Healthcare Recommendation System for Multidisciplinary Diabetes Patients with Data Fusion Based on Deep Ensemble Learning

Ihnaini et al. 2021

Smart Healthcare Recommendation System for Multidisciplinary Diabetes Patients.

Data fusion allows low-power sensors.

Data conflicts yield unexpected consequences.

12.

Novel framework based on deep learning and cloud analytics for smart patient monitoring and recommendation

Motwani et al. 2021

Categorical Cross Entropy

Helps assess model correctness

Requires more time in monitoring

13.

A remote healthcare monitoring framework for diabetes prediction using machine learning

Ramesh et. al 2021

Support Vector Machine Radial Basis Function

SVMs are good at handling high-dimensional data and small datasets.

Unsuitable to Large Datasets.

Large training time.

14.

Smart Health Monitoring System using IOT and Machine Learning Techniques

Pandey et. al 2020

Support Vector Machine

It is robust to outliers

Unsuitable to Large Datasets

15.

An IoMT-Enabled Smart Healthcare Model to Monitor Elderly

People Using Machine Learning Technique

Khan et al. 2021

Smart Healthcare Model

Fastest and most accurate medical treatment

Physical demands

17.

Cloud- and IoT-based deep learning technique-incorporated secured health monitoring system for dead diseases

Malarvizhi Kumar et. al

2021

Multi Channel Spatio-Temporal Convolutional Neural Network

Provide a reliable stock price

A lot of training data is needed

18.

Design and Development of Diabetes Management System Using Machine Learning

Sowah et. al 2020

K-Nearest Neighbour

It can naturally handle multi-class cases

It’s difficult to pick the “correct” value of K

  1. Separately write future directions at the end of the conclusion.

Answer: Thank you for your comment. Based on the comment, we have updated the manuscript.

They want to regularly gather user input in our future development and feature enhancements. It will keep our application focused on the patient, allowing us to able to take users' demands into account while refining current features and building new ones. Last but not least, we must always protect the privacy of our consumers as our first concern. The application will have openings for a data breach or leak.

  1. The security analysis of the proposed work can be included.

Answer: Thank you for your comment. Based on the comment, we have updated the manuscript.

Figure 7. Results of security of life

Providing accurate patient information to the hospital to protect the patient's life is referred to as "security of life." A threat to the patient's health might result from failure to comply. By misusing the devices, people with evil intentions might transmit inaccurate data to the hospital Figure 7 depict the outcomes of security of life. It is observed that SMO has 85%, SVM has 91%, DT has 73%, and ASV-RF has 91% in terms of security of life. This chart demonstrates that the suggested approach of ASV-RF has a high value.

  1. Make sure that proof-reading is done by the author.

Answer: Thank you for your comment. Based on the comment, we have updated the manuscript.

  1. 7Author can include more references related to the topic.

Answer: Thank you for your comment. Based on the comment, we have updated the manuscript.

The author of [22] mentions that a literature review of the work has been done, focusing on the benefits of merging telemedicine with AI. These advantages present endless growth opportunities. The article also examines how AI and telemedicine have been utilized to enhance continuous monitoring and the challenges these methods are intended to solve. The author of [23] uses the patient's glucose and blood pressure values, and the aim is to forecast their hypertension and diabetes state. Classification methods for supervised machine learning are used. In this case, a system is taught to forecast the patient's blood pressure and diabetes state. The support vector machine classification method was deemed the most accurate after evaluating all the classification algorithms and was therefore selected to train the model. The study [24] provides a model that uses light transmission to calculate the amount of glucose in the body. As the Li-Fi technology is both quicker and more efficient than the conventional Wi-Fi networks, it is the one that is employed. The author of [25] mentions that IoT with AI is used to examine the healthcare sectors to improve patient assistance and patient care in the future direction. Traditional healthcare assistance systems fail to predict the exact patient health information and needs, reducing the accuracy of the patient assistance process. An Internet of Things sensor equipped with artificial intelligence is used to accurately predict a patient's specific details, such as their fitness tracker, medical reports, health activity, body mass, temperature, and other health care information, which assists in selecting the appropriate assistance process.

References

  1. Kadu, A. and Singh, M., 2021, October. Comparative analysis of e-health care telemedicine system based on Internet of Medical Things and artificial intelligence. In 2021 2nd International Conference on Smart Electronics and Communication (ICOSEC) (pp. 1768-1775). IEEE.
  2. Chatrati, S.P., Hossain, G., Goyal, A., Bhan, A., Bhattacharya, S., Gaurav, D. and Tiwari, S.M., 2022. Smart home health monitoring system for predicting type 2 diabetes and hypertension. Journal of King Saud University-Computer and Information Sciences, 34(3), pp.862-870.
  3. Tirkey, A. and Jesudoss, A., 2020, July. A non-invasive health monitoring system for diabetic patients. In 2020 International Conference on Communication and Signal Processing (ICCSP) (pp. 1065-1067). IEEE.
  4. Fouad, H., Hassanein, A.S., Soliman, A.M. and Al-Feel, H., 2020. Analyzing patient health information based on IoT sensor with AI for improving patient assistance in the future direction. Measurement, 159, p.107757.

Reviewer 2 Report

1.      Some acronyms are with uppercase first letters of their full name definitions while some others are not. Please double-check.

2.      The abstract section should be polished. We generally only need about 150 to 200 words here to briefly introduce the background, main problem, our method and the main results from our study here.

3.      There are some other literatures about the 5G IoT-based e-health system, which should be included and discussed to highlight the difference from prior studies, for instance, https://ieeexplore.ieee.org/document/9566565, https://ieeexplore.ieee.org/document/9972679.

4.      The authors may want to highlight the original idea and contribution from their study in the revised version. It seems to the reviewer that most of the contents and technologies are some well-known ones.

5.      The author should give more detailed information about the simulation parameters and settings for a better reproduction about their results.

6.      The authors should double-check the reference format of Sensors Journal.

Author Response

Reviewer#2

  1. Some acronyms are with uppercase first letters of their full name definitions while some others are not. Please double-check.

      Thank you for the comment. Based on the above comment manuscript is       updated.

Artificial intelligence (AI), Support vector machine (SVM), Random Forest (RF), Sequential minimal optimization (SMO).

  1. The abstract section should be polished. We generally only need about 150 to 200 words here to briefly introduce the background, main problem, our method and the main results from our study here.

Answer: Thank you for your comment. Based on the comment, we have updated the manuscript.

Background: Continuous surveillance helps persons with diabetes live better lives. A wide range of technologies, such as the Internet of Things (IoT), modern communications, and artificial intelligence (AI), can assist in lowering the expense of health services. Due to numerous communication systems, it is now possible to provide customized and distant healthcare. Main problem: Healthcare data grows daily, making storage and processing challenging. We provide intelligent healthcare structures for smart e-health apps to solve the aforesaid problem. The 5G network must offer advanced healthcare services to meet important requirements like large bandwidth and excellent energy efficiency. Methodology: This research presents an intelligent system for diabetic patient tracking using machine learning (ML). The architectural components comprised smart devices, sensors, and smartphones to gather body measurements. Then, the preprocessed data is normalized using the normalization procedure. To extract features, we use Linear Discriminant Analysis (LDA). To establish a diagnosis, the intelligent system conducted data classification utilizing the suggested Advanced Spatial Vector based Random Forest (ASV-RF) in conjunction with Particle Swarm Optimization (PSO). Results: Compared to other techniques, the results of the simulation show that the suggested approach offers greater accuracy. 

Keywords: e-health, Internet of Things (IoT), machine learning (ML), diabetic patient monitoring, Advanced Spatial Vector based Random Forest (ASV-RF), Particle Swarm Optimization (PSO)

  1. There are some other literatures about the 5G IoT-based e-health system, which should be included and discussed to highlight the difference from prior studies, for instance, https://ieeexplore.ieee.org/document/9566565, https://ieeexplore.ieee.org/document/9972679.

Thank you for the comment. Based on the comment manuscript is updated.

The study [26] introduced Grey Filter Bayesian Convolution Neural Network (GFB-CNN), a real-time data-driven Deep Neural Network-driven IoT smart healthcare strategy.  They proposed a GFB-CNN-based, AI-driven Internet of Things (IoT) eHealth architecture to enhance precision and efficiency across the essential quality of service parameters. The article [27] begins with a discussion of the technologies involved in the design of 5G e-health systems from the physical layer, the application layer, and the cross-layer viewpoint.

References

  1. Suganyadevi S, Priya SS, Menaha R, Sathiya S, Jha P. Smart Healthcare in IoT using Convolutional Based Cyber Physical System. In2022 IEEE 2nd Mysore Sub Section International Conference (MysuruCon) 2022 Oct 16 (pp. 1-6). IEEE.
  2. Zhang D, Rodrigues JJ, Zhai Y, Sato T. Design and implementation of 5G e-health systems: Technologies, use cases, and future challenges. IEEE Communications Magazine. 2021 Sep;59(9):80-5.
  1. The authors may want to highlight the original idea and contribution from their study in the revised version. It seems to the reviewer that most of the contents and technologies are some well-known ones.

Answer: Thank you for your comment. Based on the comment, we have updated the manuscript.

  • The results of this study show an intelligent method for monitoring diabetes patients that uses machine learning (ML).
  • The architectural components include smart gadgets, sensors, and cellphones, all used to obtain body measurements.
  • The normalisation approach is then used to normalise the pre-processed data. Linear Discriminant Analysis is used to extract features (LDA).
  • The intelligent system performed data categorization using the proposed Advanced Spatial Vector based Random Forest (ASV-RF) together with Particle Swarm Optimization (PSO) to generate a diagnosis.
  1. The author should give more detailed information about the simulation parameters and settings for a better reproduction about their results.

Thank you for the comment. Based on the comment manuscript is updated.

The accuracy is calculated using equation (11). It's a measure of how many samples are correctly categorized. It determines the degree of similarity between the final results and the input data. The graph demonstrates how the new technique is more accurate than the old one.

                                                                                  (11)

One of the most crucial metrics for accuracy is precision, calculated as the proportion of properly classified cases to all instances of predictively positive data, as shown in equation (12). It measures the precision of the recommended Procedure by comparing the number of actual successes with the number of expected successes. The performance of the suggested technique is evaluated by distinguishing between true and false positives.

                                                                                              (12)

The proportion between the value of TNs and the total amount of TNs and FPs is referred to as specificity (equation (16)). Specificity is the likelihood of a negative outcome under the premise that the result is, in fact, negative. This probability is often referred to as the real negative rate.

                                                                                          (16)

  1. The authors should double-check the reference format of Sensors Journal.

Thank you for the comment. Based on the comment manuscript is updated.

  1. Krishna PV, Gurumoorthy S, Obaidat MS, Monisha K, Rajasekhara Babu M. A novel framework for healthcare monitoring system through cyber-physical system. Internet of things and personalized healthcare systems. 2019:21-36.
  2. Rghioui A, Lloret J, Parra L, Sendra S, Oumnad A. Glucose data classification for diabetic patient monitoring. Applied Sciences. 2019 Oct 21;9(20):4459.
  3. Alazzam MB, Mansour H, Alassery F, Almulihi A. Machine learning implementation of a diabetic patient monitoring system using interactive E-app. Computational Intelligence and Neuroscience. 2021 Dec 31;2021.
  4. Malasinghe LP, Ramzan N, Dahal K. Remote patient monitoring: a comprehensive study. Journal of Ambient Intelligence and Humanized Computing. 2019 Jan 29;10:57-76.
  5. Li X, Dunn J, Salins D, Zhou G, Zhou W, Schüssler-Fiorenza Rose SM, Perelman D, Colbert E, Runge R, Rego S, Sonecha R. Digital health: tracking physiomes and activity using wearable biosensors reveals useful health-related information. PLoS biology. 2017 Jan 12;15(1):e2001402.
  6. Rghioui A, Lloret J, Harane M, Oumnad A. A smart glucose monitoring system for diabetic patient. Electronics. 2020 Apr 22;9(4):678.
  7. Ruffini M. Multidimensional convergence in future 5G networks. Journal of Lightwave Technology. 2017 Feb 1;35(3):535-49.
  8. Mshali H, Lemlouma T, Magoni D. Adaptive monitoring system for e-health smart homes. Pervasive and Mobile Computing. 2018 Jan 1;43:1-9.
  9. Aski VJ, Dhaka VS, Kumar S, Parashar A. IoT Enabled Elderly Monitoring System and the Role of Privacy Preservation Frameworks in e-health Applications. InIntelligent Data Communication Technologies and Internet of Things: Proceedings of ICICI 2021 2022 Feb 28 (pp. 991-1006). Singapore: Springer Nature Singapore.
  10. Latchoumi TP, Dayanika J, Archana G. A comparative study of machine learning algorithms using quick-witted diabetic prevention. Annals of the Romanian Society for Cell Biology. 2021 Apr 13:4249-59.
  11. Ihnaini B, Khan MA, Khan TA, Abbas S, Daoud MS, Ahmad M, Khan MA. A smart healthcare recommendation system for multidisciplinary diabetes patients with data fusion based on deep ensemble learning. Computational Intelligence and Neuroscience. 2021;2021.
  12. Motwani A, Shukla PK, Pawar M. Novel framework based on deep learning and cloud analytics for smart patient monitoring and recommendation (SPMR). Journal of Ambient Intelligence and Humanized Computing. 2021 Jan 2:1-6.
  13. Ramesh J, Aburukba R, Sagahyroon A. A remote healthcare monitoring framework for diabetes prediction using machine learning. Healthcare Technology Letters. 2021 Jun;8(3):45-57.
  14. Pandey H, Prabha S. Smart health monitoring system using IOT and machine learning techniques. In2020 sixth international conference on bio signals, images, and instrumentation (ICBSII) 2020 Feb 27 (pp. 1-4). IEEE.
  15. Khan MF, Ghazal TM, Said RA, Fatima A, Abbas S, Khan MA, Issa GF, Ahmad M, Khan MA. An iomt-enabled smart healthcare model to monitor elderly people using machine learning technique. Computational Intelligence and Neuroscience. 2021 Nov 25;2021.
  16. Chaki J, Ganesh ST, Cidham SK, Theertan SA. Machine learning and artificial intelligence based Diabetes Mellitus detection and self-management: A systematic review. Journal of King Saud University-Computer and Information Sciences. 2022 Jun 1;34(6):3204-25.
  17. Malarvizhi Kumar P, Hong CS, Chandra Babu G, Selvaraj J, Gandhi UD. Cloud-and IoT-based deep learning technique-incorporated secured health monitoring system for dead diseases. Soft Computing. 2021 Sep;25(18):12159-74.
  18. Sowah RA, Bampoe-Addo AA, Armoo SK, Saalia FK, Gatsi F, Sarkodie-Mensah B. Design and development of diabetes management system using machine learning. International journal of telemedicine and applications. 2020 Jul 16;2020.
  19. Rghioui, A., Naja, A., Mauri, J.L. and Oumnad, A., 2021. An IoT based diabetic patient monitoring system using machine learning and node MCU. In Journal of Physics: Conference Series (Vol. 1743, No. 1, p. 012035). IOP Publishing.
  20. Rghioui A, Lloret J, Sendra S, Oumnad A. A smart architecture for diabetic patient monitoring using machine learning algorithms. InHealthcare 2020 Sep 19 (Vol. 8, No. 3, p. 348). MDPI.Godi B, Viswanadham S, Muttipati AS, Samantray OP, Gadiraju SR. E-healthcare monitoring system using IoT with machine learning approaches. In2020 international conference on computer science, engineering and applications (ICCSEA) 2020 Mar 13 (pp. 1-5). IEEE.
  21. Rghioui A, Lloret J, Sendra S, Oumnad A. A smart architecture for diabetic patient monitoring using machine learning algorithms. InHealthcare 2020 Sep 19 (Vol. 8, No. 3, p. 348). MDPI.
  22. Godi B, Viswanadham S, Muttipati AS, Samantray OP, Gadiraju SR. E-healthcare monitoring system using IoT with machine learning approaches. In2020 international conference on computer science, engineering and applications (ICCSEA) 2020 Mar 13 (pp. 1-5). IEEE. 

Round 2

Reviewer 2 Report

No further comments.